# Measurements of Hydrodynamics, Sediment, Morphology and Benthos on Ameland Ebb-Tidal Delta and Lower Shoreface

Bram C. van Prooijen[1], Marion F.S. Tissier[1], Floris P. de Wit[1], Stuart G. Pearson[1,3], Laura B. Brakenhoff[2], Marcel C.G. van Maarseveen[2], Maarten van der Vegt[2], Jan-Willem Mol[6], Frank Kok[6], Harriette Holzhauer[4,3], Jebbe J. van der Werf[3,4], Tommer Vermaas[3], Matthijs Gawehn[3], Bart Grasmeijer[3], Edwin P.L. Elias[3], Pieter Koen Tonnon[3], Giorgio Santinelli[3], Jose A.A. Antolinez[1], Paul L.M. de Vet[1,3], Ad J.H.M. Reniers[1], Zheng Bing Wang[1,3], Cornelis den Heijer[1,5], Carola van Gelder-Maas[6], Rinse J.A. Wilmink[6], Cor A. Schipper[6], and Harry de Looff[6]

[1]Delft University of Technology, Delft, the Netherlands
[2]Utrecht University, Utrecht, the Netherlands
[3]Deltares, Delft, the Netherlands
[4]University of Twente, Enschede, the Netherlands
[5]Data2day, Delft, the Netherlands
[6]Rijkswaterstaat, Lelystad, the Netherlands

**Correspondence:** B.C. van Prooijen (B.C.vanProoijen@TUDelft.nl)

**Abstract.** A large-scale field campaign has been carried out on the ebb-tidal delta (ETD) of Ameland Inlet, a basin of the Wadden Sea in the Netherlands, as well as on three transects along the Dutch lower shoreface. The data has been obtained over the years 2017-2018. The most intensive campaign at the ETD of Ameland Inlet was in September 2017.

With this campaign, as part of KustGenese2.0 (Coastal Genesis 2.0) and SEAWAD, we aim to gain new knowledge on the processes driving sediment transport and benthic species distribution in such a dynamic environment. These new insights will ultimately help the development of optimal strategies to nourish the Dutch coastal zone in order to prevent coastal erosion and keep up with sea level rise. The dataset obtained from the field campaign consists of: (i) single and multi-beam bathymetry; (ii) pressure, water velocity, wave statistics, turbidity, conductivity, temperature, and bedform morphology on the shoal; (iii) pressure and velocity at six back-barrier locations; (iv) bed composition and macro benthic species from box-cores and vibrocores; (v) discharge measurements through the inlet; (vi) depth and velocity from X-band radar; and (vii) meteorological data.

The combination of all these measurements at the same time makes this dataset unique and enables us to investigate the interactions between sediment transport, hydrodynamics, morphology and the benthic ecosystem in more detail. The data provides opportunities to calibrate numerical models to a high level of detail. Furthermore, the open-source data sets can be used for system comparison studies.

The data is publicly available at 4TU Centre for Research Data at https://doi.org/10.4121/collection:seawad (Delft University of Technology et al., 2019) and https://doi.org/10.4121/collection:kustgenese2 (Rijkswaterstaat and Deltares, 2019). The data sets are published in netCDF format and follow conventions for CF (Climate and Forecast) metadata. The data.4tu.nl site provides keyword searching options and maps with the geographical position of the data.

# 1 Introduction

Systems of barrier islands and associated tidal inlets are found along a major part of the world's coastlines (Glaeser (1978); Stutz and Pilkey (2011)). The sheltered back-barrier basins have been attractive for human settlement and all kinds of recreational and economic activities. The inlet systems form unique landscapes with channels, shoals and salt marshes, providing

valuable habitats for numerous marine species and birds. Over the last decades, these systems have been under increasing pressure due to economic activities, while also the awareness of the uniqueness of the areas has been recognized and formalized. For example, the Wadden Sea (the Netherlands, Germany and Denmark) was identified as World Heritage in 2009 for its unique geological and ecological values. Accounting for all the functions and values of the system requires a careful sustainable management strategy. This is even more necessary in view of the foreseen climate change. Relative sea level rise threatens

these systems in various ways. The higher water level reduces the safety level of the barrier islands, while the intertidal flats in the basins might risk drowning if they do not keep pace with rising waters (Wang et al., 2018).

To keep the safety standards for the Dutch coast, the coastal zone is being nourished. The strategy is to let the sediment volume in the coastal zone (as defined in Figure 1) keep pace with sea level rise Mulder et al. (2011). The coastal zone may however lose sediment over the offshore boundary (approximately the depth contour MSL -20m) and through the inlets towards

the Wadden Sea. Quantifying these fluxes is not trivial (Van Rijn, 1997). The net fluxes into the Wadden Sea depend on subtle variations in the large flood and ebb fluxes (Gatto et al., 2017). Furthermore the dynamics of the ebb tidal delta make predictions complicated. The flux over the offshore boundary at the lower shoreface is complicated by the episodic nature of the sediment transport at this depth.

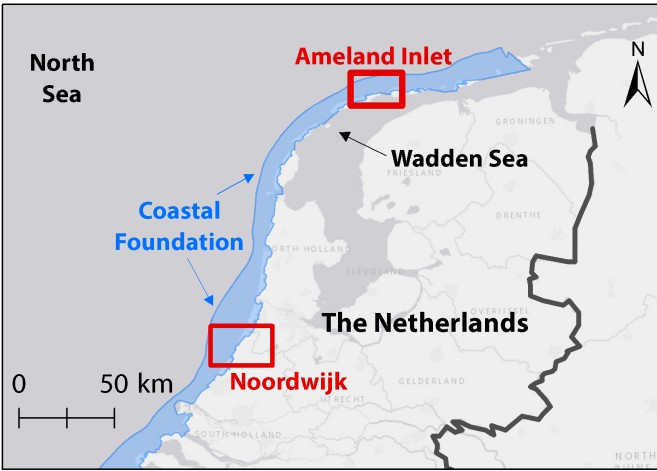

**Figure 1.** Location of Ameland Inlet and the Wadden Sea within the Netherlands. Basemap sources: Esri, HERE, Garmin, ©OpenStreetMap contributors, and the GIS user community.

Numerous conceptual models have been formulated to explain sediment dynamics and interactions at barrier island type inlets (see a recent summary by Hayes and FitzGerald (2013)). These conceptual descriptions provide a general understanding of the factors controlling the shape, size and general characteristics of a tidal inlet and its associated ebb-tidal delta. The size of the inlet is primarily controlled by the tidal prism, e.g. (Jarrett, 1976), while the ratio between wave and tidal energy controls the geometry of the barrier islands, the inlet gorge and the ebb-tidal delta. Such a conceptual understanding of large-scale behavior is a crucial first step. However, a more detailed and quantitative description is required to answer current predictive needs. Furthermore, these conceptual models often lack a description of the underlying physical processes. Knowledge on these processes is essential if one aims to understand changes on smaller scales, where human intervention may influence the behavior such that it cannot be accurately described by existing concepts and equilibrium relationships. However, process-based models require accurate and high-resolution data for calibration and validation. Suitable field datasets that comprise sufficient and coherent observations of hydrodynamics, sediment transport and morphological change are scarce, as tidal inlets are notoriously challenging and expensive places to collect field data.

The Dutch Government Rijkswaterstaat therefore started the KustGenese2.0 research program in collaboration with Deltares. Part of this program was an extensive field campaign at the Ameland Inlet, the Netherlands (Figure 1), in close collaboration with the universities of Delft, Utrecht and Twente, via the SEAWAD project. Hydrodynamics, turbidity, sediment composition and benthic species distribution were measured at various locations on the ebb-tidal delta, in the inlet gorge and in the basin. Additionally, measurements were carried out at the Holland Coast, near Noordwijk. These datasets help us to (1) improve our understanding of the physical processes underlying mixed-energy tidal inlets, (2) formulate new algorithms describing these physics, and (3) evaluate the skill of process-based numerical models, and if necessary improve the underlying model formulations. Ultimately, the obtained insights and improved models will lead to more efficient and effective management of the barrier island system of the Wadden Sea to prevent coastal erosion and keep up with sea level rise.

This paper describes the datasets obtained in 2017 and 2018. The dataset is accessible via https://doi.org/10.4121/collection: seawad and https://doi.org/10.4121/collection:kustgenese2. The repositories include the raw and processed data as well as relevant metadata and processing scripts.

## 2 Area Description

The Wadden Sea (Figure 1) consists of a series of 33 tidal inlet systems and in total extends over a distance of nearly 500 km along the northern part of the Netherlands (West Frisian Islands) and the North Sea coasts of Germany and Denmark (the East and North Frisian Islands). The tidal basins consist of extensive intertidal areas and tidal channels that support a wide variety of marine mammals, birds and fish. This unique natural habitat was selected as a world heritage site in 2009. Ameland Inlet is centrally located in the Dutch part of the Wadden Sea, bordered by the islands Terschelling to the west and Ameland to the east (Figure 1 1). The associated Ameland tidal basin has a length of about 30 km and covers an area of around 309 km$^2$. With a tidal range of approximately 2 m and a moderate wave climate, the inlet can be classified as meso-tidal and mixed-energy (Hayes, 1975; Davis and Hayes, 1984).

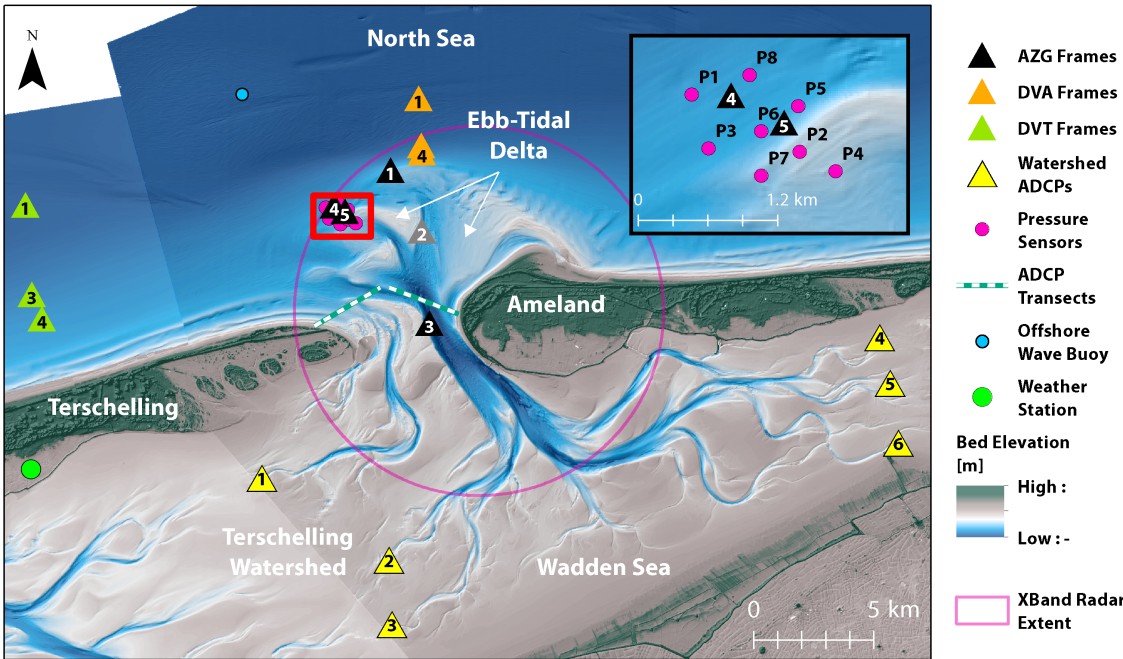

**Figure 2.** Locations of hydrodynamic, sediment, and atmospheric measurements carried out during the 2017 campaigns. (AZG = *Amelander Zeegat* or Ameland Inlet; DVA = *Diepere Vooroever Ameland* or Lower Shoreface Ameland; DVT = *Diepere Vooroever Terschelling* or Lower Shoreface Terschelling). Bathymetry source: Rijkswaterstaat Vaklodingen. Elevation source: Actueel Hoogtebestand Nederland (AHN), Rijkswaterstaat.

Ameland Inlet is considered to be relatively undisturbed as no major ongoing human interferences or interventions in the past directly impact the natural processes. However, Elias et al. (2012) points out that natural processes in the Wadden Sea can only reign free within its established boundaries. Over the last centuries, multiple large- and small-scale interventions, such as coastal defence works, closure dams, dikes, sea-walls, and land reclamations, closing of the Middelzee around 1600, have reduced and essentially fixed the basin dimensions and kept the barrier islands in place. As a result, a geomorphic transition in morphodynamic behavior of Ameland Inlet occurred around 1926 as the main ebb-channel migrated from an updrift to a downdrift position in the inlet gorge, encroaching on the western side of Ameland (Elias et al., 2019). This channel has retained this position since then, partly due to extensive coastal protection works at the tip of the island. Within this context, natural processes can reshape the individual shoals and channels on the ebb-tidal delta, without human interference.

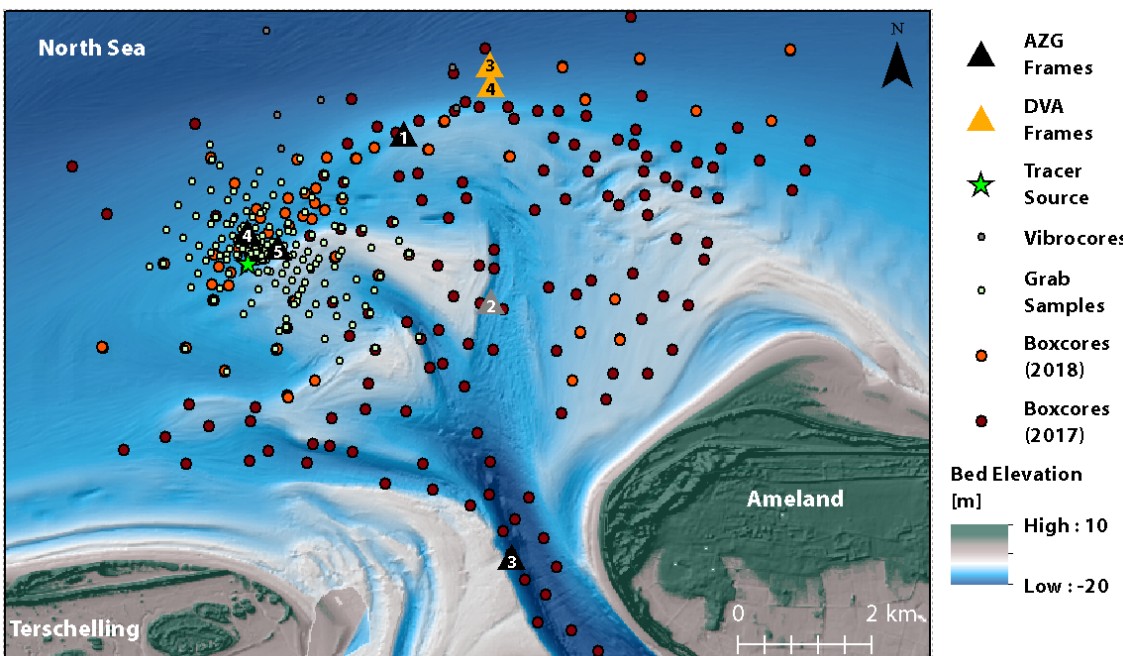

**Figure 3.** Locations of seabed sediment samples taken in September and October 2017, including boxcores, van Veen grab samples, and vibrocores. The source location for the sediment tracer study is indicated with a green star. Measurement frames from the AZG (Ameland Inlet) and DVA (Ameland Foreshore) campaigns are shown as triangles for context. Bathymetry source: Rijkswaterstaat Vaklodingen. Elevation source: Actueel Hoogtebestand Nederland (AHN), Rijkswaterstaat.

## 3 Instrumentation

### 3.1 Frames

Five frames were built and equipped with: Acoustic Doppler Velocimeters (ADV), an upward-looking Acoustic Doppler Current Profiler (ADCP), a downward-looking high-resolution ADCP, Optical Backscatter sensors (OBS), a Laser In-Situ Scattering and Transmissometery (LISST) sensor, a Sonar and a Multi-Parameter Probe. The exact configuration slightly differs per frame. As an example, Figure 4 and Table 1 describe the configuration for Frame 4.

The procedure for determining the offset of the compass was to rotate the mounting frame annotating every ten degrees the device heading angle (compass heading) and the true angle measured with high accuracy GPS (magnetic heading) not affected by the frame. This was repeated in reverse direction. An averaged compass deviation (from the two cycles) at a 10-degree interval was taken for the compass calibration.

High frequency measurements offer possibilities to analyse intra-wave processes and turbulence characteristics. The **ADV** sampled with a frequency of 16Hz (Nortek Vector) or 10Hz (Sontek Hydra) in almost continuous mode: 29 minute bursts were measured at an interval of 30 minutes. The instruments measured the distance to the bed level at the beginning and end of the

bursts, providing timeseries of the bed level every 30 minutes. Generally, at least two ADVs were placed near the seabed to analyse flow properties at different heights (ranging from 0.35 to 1.00 m above the bed).

The velocity profile over the full water column was measured to determine the tidal flow and wave orbital motion. The **upward-looking ADCP** (Teledyne RDI Workhorse Monitor) measured with a frequency of 1.25 Hz with bursts of 30 minutes at intervals of 30 or 60 minutes. The cell sizes differed for the various deployment locations: 0.25, 0.50, 0.80, or 1 m. The number of cells was always sufficient to cover the full water column above the instrument.

To measure the near-bed profile, including the wave orbital motions, a high resolution ADCP was deployed. The **downward-looking ADCP** (Nortek Aquadopp HR) measured with 4 Hz in near-continuous mode: 29 min bursts were measured over an interval of 30 minutes. The cell size was set to 0.03 m for 13 cells, providing a maximum representative profile height of 0.39 m.

Turbidity was measured to determine the sediment concentration. Four **OBSes** (Campbell OBS-3+ Turbidity Sensor) were attached to a leg of each frame, close to the bed (0.1-0.8 m). They were each connected to an ADV and measured with the same frequency, burst interval and period. A calibration was carried out at the laboratory of Utrecht University. Sediment (mainly sand) from the seabed was taken and mixed up in a mixing tank. The sediment concentration was slowly increased, obtaining the relation between voltage and sediment concentration. As the suspended sediment in the field also contained fines, the results from the laboratory (containing sand) cannot be translated directly into a mass concentration (Pearson et al., 2019). Therefore, only the raw voltage output is provided.

A **LISST** (LISST-100X Particle Size Analyzers, Sequoia) uses the scattering and transmission of a laser to analyse the suspended particle size distribution at a point 0.6 m above the bed. Every 60 seconds, there is a 15 second continuous burst of measurements at 1 Hz. The instrument can distinguish particles within a range of 2.5 $\mu$m - 500 $\mu$m.

The **3D Sonar** measures the detailed bed morphology and can be used to detect small-scale bedforms like ripples. The 3D Sonar (Marine Electonics type 2001) was mounted approximately 1 m above the bed. It measured with intervals of 1 hour by rotating around its own axis and scanning 200 swaths of the bed in a 360° circle, thus all swaths are 0.9° apart. The maximum deviation of the Sonar beam with the vertical was 75°, and the resolution within the swaths was 0.9°. The horizontal resolution is highest right below the Sonar head (1.6 cm) and decreases to 22 cm at the edges of the swath (>3.5 m away from the centre of the image). The vertical resolution is approximately 4.5 mm.

The **Multi-Parameter Probe** measures pressure, temperature, conductivity, pH, turbidity, chlorophyll, blue-green algae phycocyanin, and optical dissolved oxygen with a frequency of 1 Hz, and was mounted 1.3 m above the seabed. These data can be used to calculate salinity and density, and to analyse the interaction between physical and biogeochemical processes. The turbidity (in NTU: Nephelometric Turbidity Unit) was a direct output of the multiprobe, applying a relation between the measured voltage and turbidity, provided by the manufacturer.

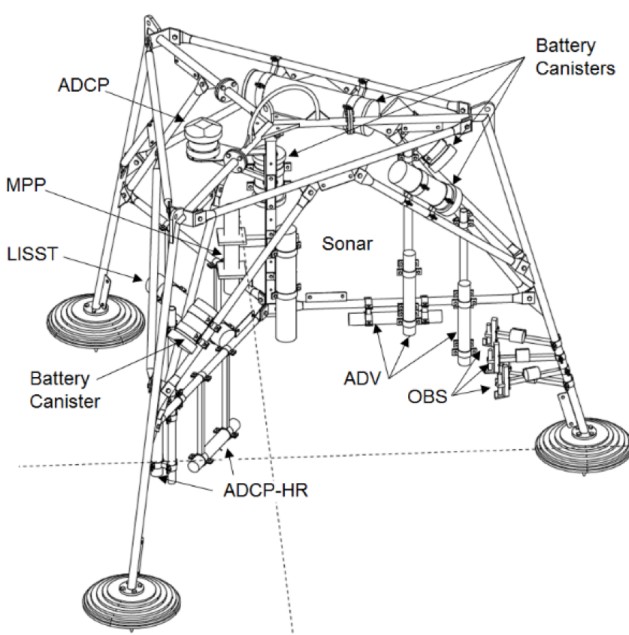

**Figure 4.** Configuration of the instruments on one of the 2.4m-high frames (Frame 4). See main text for an explanation for each instrument and Table 1 for a summary of the instruments' settings.

## 3.2 Wave measurements

Offshore wave conditions were measured by a Datawell Directional Waverider Mk3 (indicated by the light blue dot in Figure 2). It provides half hourly estimates of spectral wave characteristics (incl. significant wave height, mean and peak period and mean direction) during the field campaign.

5      To further characterise the spatial variability in wave characteristics at the Ameland ebb-tidal delta, eight additional **pressure sensors** (Ocean Sensor Systems, Inc.) were installed 25-50 cm above the seabed around Frames 4 and 5 (P1-P8, see magenta dots in Figure 2). 10-Hz continuous pressure data was successfully retrieved from seven of these instruments (all but P6). These pressure signals were subsequently corrected for variations in the atmospheric pressure (see also Section 3.1). They can be used to characterise mean water level and wave statistics at the ebb-tidal delta, but also intra-wave properties.

10   ## 3.3 Discharge and Velocity Measurements

Transect measurements were carried out at two transects in the opening between the islands Terschelling and Ameland (see dashed lines in Figure 2). The velocities in the full cross section were measured by research vessels equipped with a downward-looking ADCP moving along the transect over a period of at least 13 hours to cover a full tidal cycle. Water samples were taken during the measurements, in order to convert backscatter to SPM concentrations later on.

| Frame 4, August 29th 2017 - October 9th 2017 | | |
|---|---|---|
| Instrument | Height above bed | Settings |
| Upward ADCP | 2.30 m | 30 min bursts every 60 min, bin size 0.25 m, 1.25 Hz |
| Downward ADCP | 0.52 m | 29 min bursts every 30 min, bin size 0.03 m, 4 Hz |
| Low ADV | 0.36 m | 29 min bursts every 30 min, 16 Hz |
| Mid. ADV | 0.65 m | 29 min bursts every 30 min, 16 Hz |
| High ADV | 0.93 m | 29 min bursts every 30 min, 16 Hz |
| Low OBS | 0.19 m | 29 min bursts every 30 min, 16 Hz |
| Middle OBS | 0.30 m | 29 min bursts every 30 min, 16 Hz |
| High OBS | 0.50 m | 29 min bursts every 30 min, 16 Hz |
| Highest OBS | 0.79 m | 29 min bursts every 30 min, 16 Hz |
| LISST | 0.60 m | 15 sec bursts every 60 sec, 1 Hz |
| MPP | 1.27 m | 1 sample every 5 min |
| Sonar | 0.98 m | 1 3D image every hour |

**Table 1.** Instrumentation for Frame 4 during the AZG deployment. The number of instruments and their settings varied by frame and by deployment. For a complete overview, see repository.

Flow measurements were carried out at the end of the tidal channels in the back-barrier basins, close to the border with the adjacent basin, see Figure 2. These borders are referred to as tidal divides (van Veen et al., 2005). These locations are chosen to determine the flow between the between the basins. At each tidal divide, three upward-looking ADCPs (Nortek Aquadopp LR) were placed, see yellow triangles in Figure 2. These measured with an interval of 1 minute. The cell size was set to 0.10 m for 45 cells. This implies that the full water column was always covered.

The measurements in the opening and at the tidal divides can be used to analyse the water budgets of the Ameland Basin.

### 3.4 Bathymetry

Half-yearly bathymetric surveys of the ebb-tidal delta were conducted between 2016 and 2019. These datasets are an addition to the regular bathymetric monitoring conducted here and follow similar protocols. Ameland Inlet has a long history of bathymetric surveying (Elias et al., 2019). Since 1985, bathymetric data are collected systematically by Rijkswaterstaat, which is part of the Ministry of Public works and Infrastructure, following the *Vaklodingen* protocol (De Kruif, 2001). More specifically, the ebb-tidal delta is measured with approximately 200 m transect spacing using a single-beam echo-sounder. Following quality checks for measurement errors, data are reduced to 1 m transect resolution, combined with nearshore coastline measurements and Lidar data for the tidal flats in the basin, and interpolated to 20x20 m grids. The grids are stored digitally as 10x12.5 km blocks called *Vaklodingen*.

In addition to the ebb-delta scale maps, detailed multi-beam echo sounding surveys were conducted at four focus areas at several intervals. The raw data were cleaned for data outliers, sub-gridded to 1 m resolution and mosaicked in single datasets. The high resolution renderings allow us to visualise and analyse bedforms characteristics such as height, asymmetry and migration. Assuming that the bedforms are still active and governed by present-day hydrodynamic conditions, the bedform distribution, arrangement and morphology provides information about the locally dominant bottom currents and sediment transport (Boothroyd, 1985; Fraccascia et al., 2016).

The Navigational X-Band radar (Terma Scanter 2000 with VV polarization) on the lighthouse of Ameland was used as a remote sensing tool to estimate depths and currents in the outer delta. The area that is captured by the radar covers a circle with a radius of approximately 7.5 km, see Figure 1. Video fragments of 12 min were stored every 20 minutes, with a frame rate of 1 image per 2.85 seconds.

## 3.5  Seabed composition and benthos

Bed samples were taken with a Reineck boxcorer (0.078 m$^2$). A map of the sampling locations is shown in Figure 3. Sediment samples were taken from the top 8cm of the boxcore and analysed with a Malvern Mastersizer resulting in a sediment distribution with 67 bins, ranging from 0.01 $\mu$m- 2000 $\mu$m. After sieving over a 1 mm sieve macrobenthic species were conserved and analysed in the laboratory. Additionally samples were taken with a square boxcore to create laquer peals of selected cores. Samples with a 6 m vibrocore were taken at the lower shoreface. Finally, a Van Veen grab sampler was used to sample the seabed to a depth of 5 cm from the surface at 187 locations. The grain sizes of these samples were also determined with the Malvern Mastersizer.

## 4  Deployments

The majority of the measurements were carried out in the period from August 29th until October 10th, 2017 during the so-called AZG campaign. This was the period when the five frames and the eight stand-alone pressure sensors were deployed at the Ameland ebb-tidal delta, the velocity transects were measured, the multibeam measurements were carried out, and the sediment samples (boxcores and grabs) were taken. The frame measurements covered a period of 40 days, see Figure 4. The frames were serviced after 3 weeks and redeployed at the same location. Except for one bent ADV stem, all instruments were intact. At the end of the period, four frames were retrieved without damage of instruments (Frames 1, 3, 4 and 5). Data has been retrieved for all instruments. The LISST on Frame 5 did not work properly and no usable data was obtained. Frame 2 (see grey triangle in Figure 2) was covered with sand after a storm, due to migration of the channel bank. As of this writing, it has not yet been retrieved.

Three of the frames (Frames 1, 3 and 4) were then re-deployed in the period 8 November 2017 till 11 December 2017 on a transect perpendicular to Ameland (DVA frames, Figure 2). These three frames were finally re-deployed in the periods 11 January 2018 till 6 February 2018 and 12 March 2018 till 26 March 2018 on a transect perpendicular to Terschelling (DVT frames, Figure 2). Finally, the frames were deployed offshore Noordwijk (Figure 1) from 4 April, 2018 to 15 May 2018.

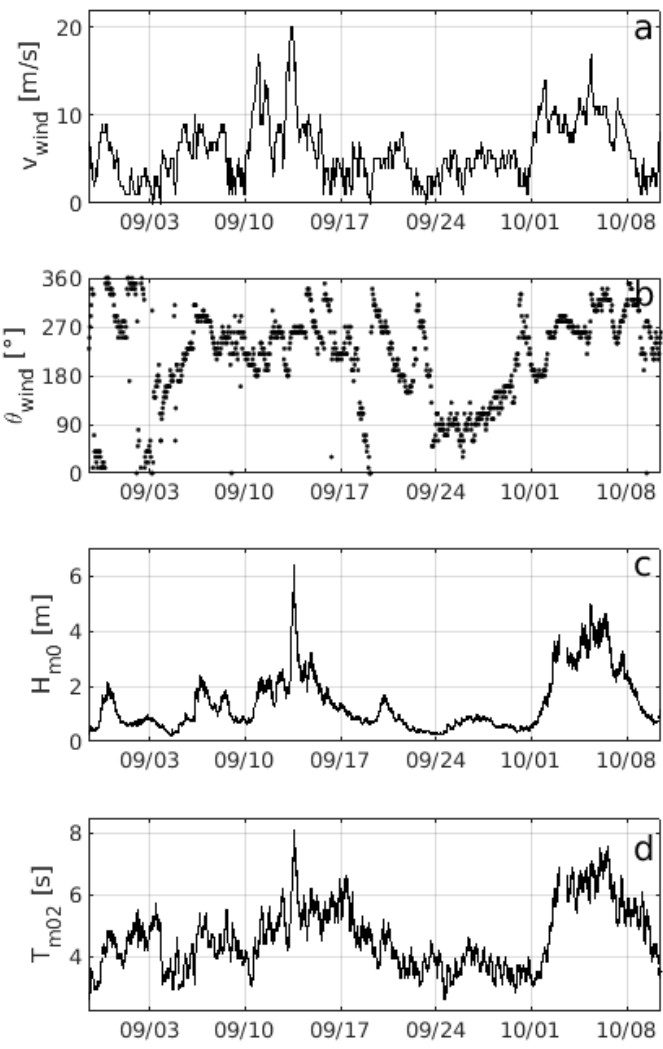

**Figure 5.** Environmental conditions for the first measurement period (29[th] of August 2017 till 10[th] of October 2017). Wind speed (a) and direction (b) measured at KNMI location 251 Hoorn Terschelling (green dot in Fig 2); Offshore significant wave height $H_s$ (c) and mean spectral period $T_{m02}$ (d) measured by wave buoy AZG-B11 (blue dot in Fig 2).

| Campaign | Frame | Begin time | End time | Lat | Lon | Approx. depth |
|----------|-------|------------|----------|-----|-----|---------------|
| | 1 | 30/08/2017 10:11 | 09/10/2017 15:20 | 53.50 ° | 5.57 ° | 8 m |
| | 2 | 30/08/2017 16:38 | N/A[1] | 53.48 ° | 5.59 ° | 9 m |
| **AZG** | 3 | 30/08/2017 15:37 | 10/10/2017 07:10 | 53.44 ° | 5.59 ° | 20 m |
| | 4 | 29/08/2017 15:55 | 09/10/2017 15:50 | 53.49 ° | 5.54 ° | 5 m |
| | 5 | 29/08/2017 15:28 | 09/10/2017 16:45 | 53.49 ° | 5.54 ° | 4 m |
| | 1 | 08/11/2017 13:00 | 11/12/2017 13:15 | 53.53 ° | 5.59 ° | 20 m |
| **DVA** | 3 | 08/11/2017 11:00 | 11/12/2017 14:15 | 53.51 ° | 5.59 ° | 16 m |
| | 4 | 08/11/2017 10:30 | 11/12/2017 15:00 | 53.51 ° | 5.59 ° | 10 m |
| | 1 | 11/01/2018 12:20 | 06/02/2018 09:30 | 53.49 ° | 5.34 ° | 20 m |
| **DVT1** | 3 | 11/01/2018 14:00 | 06/02/2018 10:30 | 53.45 ° | 5.35 ° | 14 m |
| | 4 | 11/01/2018 15:15 | 06/02/2018 11:30 | 53.45 ° | 5.35 ° | 10 m |
| | 1 | 12/03/2018 16:00 | 26/03/2018 10:10 | 53.49 ° | 5.34 ° | 20 m |
| **DVT2** | 3 | 12/03/2018 19:50 | 26/03/2018 13:40 | 53.45 ° | 5.35 ° | 14 m |
| | 4 | 12/03/2018 17:50 | 26/03/2018 12:40 | 53.45 ° | 5.35 ° | 10 m |
| | 1 | 04/04/2018 12:15 | 15/05/2018 13:30 | 52.28 ° | 4.24 ° | 20 m |
| **DVN** | 3 | 04/04/2018 14:10 | 15/05/2018 17:00 | 52.23 ° | 4.39 ° | 12 m |
| | 4 | 04/04/2018 13:40 | 15/05/2018 14:50 | 52.24 ° | 4.37 ° | 16 m |

**Table 2.** Overview of the measurement periods and positions of the frames. The campaigns are referred to by AZG (*Amelander ZeeGat* in Dutch): Amelander Inlet ; DVA (*Diepe Vooroever Ameland*): Lower Shoreface Ameland; DVT1 (*Diepe Vooroever Terschelling 1*): Lower Shoreface Terschelling 1; DVT2 (*Diepe Vooroever Terschelling 2*): Lower Shoreface Terscheling 2; and DVN (*Diepe Vooroever Noordwijk*): Lower Shoreface Noordwijk.

An overview of the deployments of the frames is provided in Table 2. Sediment composition and macrobenthic species were sampled in two surveys. During the first survey, 4-6 September 2017 and 20-21 September 2017, 166 samples were collected. The second survey with 55 samples took place on the $24^{th}$ of March 2018.

## 5  Data Processing

The ADV and ADCP data were processed in two steps. In the first step, the correlation and Signal-to-Noise-Ratio (SNR) were determined. Threshold values for correlation and SNR were based on Elgar et al. (2005). Secondly, velocities were despiked by using the 3D phase space method (Goring and Nikora, 2002; Mori et al., 2007), in which velocities and their first and second order derivatives are plotted in a 3D space. Subsequently, points outside a given ellipsoid are excluded. The flagged data are replaced by NaNs (Not a Number). Positioning and orientation of the instruments was based on a laser scan of the frame and the calibration of the compasses.

To calibrate the LISST, the background scatter intensity of the laser in clean water must be measured. This procedure was carried out prior to each campaign in accordance with the manufacturer's specifications. This calibration stage ensures that the laser detection rings are properly aligned and provides a basis for interpreting the measurements on site. Upon retrieval of the data from the instrument, the raw data was processed using the LISST-SOP Version 5.0.50 software. No despiking or filtering was carried out on the time series.

Pressure signals are measured by the ADVs, ADCPs, Aquadopps, LISST, Multi-parameter probe and standalone pressure sensors. The pressure sensors measure the total pressure, which is the combination of atmospheric pressure and water pressure. To obtain the water pressure, the total pressure is reduced by the air pressure. The air pressure is obtained from the nearest meteo station (Terschelling Hoorn AWS) of the Royal Netherlands Meteorological Institute (KNMI).

The point clouds of each Sonar scan were interpolated on a regular grid with a 0.01 m step size using a second-order LOESS interpolator, following Ruessink et al. (2015). This interpolator also removes spikes. Because of the low resolution at the edges, the grids run from -2.5 to 2.5 m in both $x$ and $y$-direction. The mean distance to the Sonar head was removed from the depth values, so larger bed level variations through time are not visible anymore. The Sonar does not store its own heading, so all images were rotated to the N-E-S-W scheme using the rotation angles of the other instruments. Data quality was checked for each image. A flag of '-1' was assigned to data should be treated with caution (e.g. if the amount of sediment suspension was too high for the Sonar to detect the bed). Data with good quality were assigned a flag '1'.

The vibrocores were subdivided in 1 m parts on board and further processed in the lab. The cores were opened and photographed, and a lithostratigraphic description of the cores was made following the 'Standard Core Description method' of the Dutch Geological Survey (Bosch et al., 2000).

Depths and currents were also estimated from the radar data using a depth-inversion algorithm called XMFit (X-Band MATLAB Fitting), see (Friedman, 2013). This algorithm is based on the fitting of the wave linear dispersion relationship on the radar-derived image intensity dispersion shell in the wave number-frequency space. The accuracy depends on the distance from the lighthouse. The system returned 1 to 3 depth and surface current estimates per hour depending on the quality of the radar backscatter.

## 6 Data Availability

The data presented in this article has been published at 4TU Centre for Research Data, see https://doi.org/10.4121/collection:seawad (Delft University of Technology et al., 2019) and https://doi.org/10.4121/collection:kustgenese2 (Rijkswaterstaat and Deltares, 2019) following the FAIR principles (Wilkinson et al., 2016). The datasets are published in netCDF format and follow conventions for CF (Climate and Forecast) metadata. The underlying raw data as produced by the instruments together with the scripts for conversion to netCDF with metadata are maintained under version control (subversion). Conversion scripts are written in Python or Matlab and developed to run platform independent. The metadata in the netCDF files specifies the date and version number of underlying raw source data and conversion script in order to provide replicability information. Each collection (SEAWAD or KUSTGENESE2) contains data sets for types of instruments (e.g. "KUSTGENESE2.0/SEAWAD Velocitymeter and

Turbidity sensor"). These datasets are subsequently split into directories per area and period (e.g. "2017_09_ameland_azg/", indicating the Amelander Inlet for September 2017). Each directory contains the netCDF files (e.g. adv_azg_201709_F3.nc containing the ADV results for Ameland Inlet for September 2017, for Frame 3). The data could also be found by searching with the keywords "Ameland and ADV". An interactive map is available, indicating the measurement locations for each dataset.

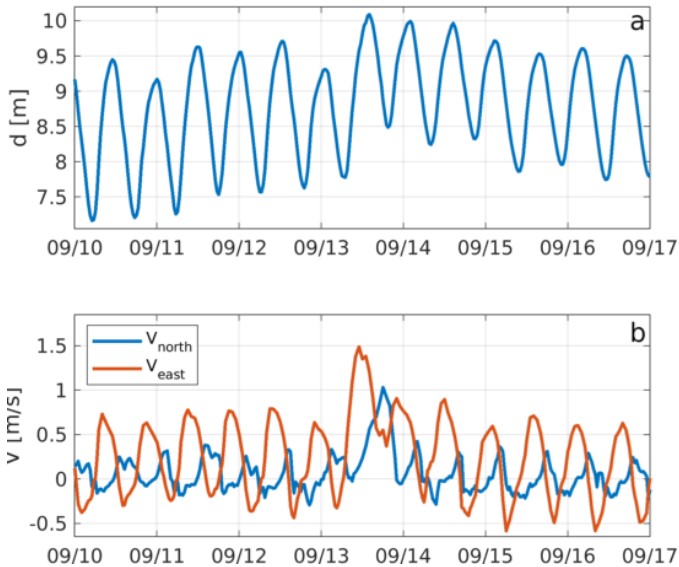

**Figure 6.** Results from Frame 4 for the period 10th of September 2017 till 17th of September 2017. (a) Mean water depth; (b) Depth-averaged mean flow velocity estimated from the upward looking ADCPs.

## 7 Environmental Conditions

### 7.1 Hydrodynamics

The wind speed and direction as well as the offshore significant wave height and mean spectral period are shown in Figure 5 for the first part of the field campaign. During this part of the campaign, two storms occurred: Aileen/Sebastian (11-13 September) and Xavier (3-5 October). A wind speed up till 20 m/s (13th of September) was measured during Sebastian. A calm weather period (wind speed <8 m/s) was present during 16-29 September. During the other deployment periods, several storms were captured as well, like the one on the 18th of January. The high wind speed on the 13th of September resulted in significant water level set up (Figure 6a) and significant wind-induced flows (Figure 6b) at the ebb-tidal shoal (Frame 4). This shallow area with a relatively steep bed slope is very sensitive to meteorological conditions: there is a strong interaction between tidal flow, wind- and wave-driven flow, and waves. This makes the location suitable for analysis of these types of hydrodynamic

interactions. The flow velocity at Frame 3 (not shown) is significantly different, as it is located on the side of the deep inlet channel. Wave-induced currents are of minor importance and the flow is highly tide dominant at this location. Storm surges on the North Sea do lead to variations in water level and discharge through the inlet, but the variation due to spring-neap tide variations is more significant. Inside the basin, the wind does have a stronger effect again. Especially storms from the southwest

(the dominant wind direction) lead to increased flow velocities on the tidal divides inside the basin. During various periods of southwestern wind, the flow is dominated by the wind, leading to unidirectional flow over a full tide, i.e. the tidal forcing is not strong enough to reverse the flow.

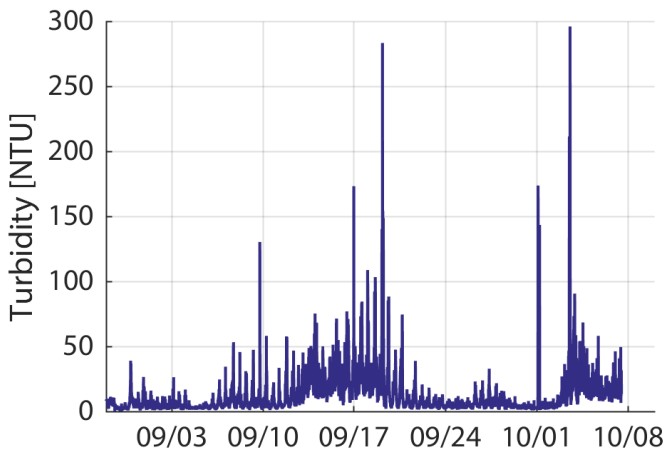

**Figure 7.** Turbidity measured at AZG Frame 4 (8 m depth) on the ebb-tidal delta using the YSI multiparameter turbidity probe, located 1.3 m above the seabed. Peaks in turbidity correspond to the two major storms, but in calmer conditions the water can also become turbid at low water slack, when suspended matter is ejected from the Wadden Sea.

## 7.2 Suspended sediment

To assess sediment transport in the Ameland Inlet system, suspended particle concentrations and turbidity were measured.

Turbidity at the distal end of the ebb-tidal delta generally shows an increase in response to energetic conditions (i.e. storms), and also to calmer conditions near low water slack (Figure 7). Pearson et al. (2019) attribute these differences to locally resuspended fine sand ($d_{50} \simeq 210\mu$m) or flocculated fine sediment and organic particles advected from the Wadden Sea.

Measurements of suspended particle size distribution and volumetric concentration ($\mu L/L$) were also obtained using LIS-STs, which showed similar responses to the YSI multiprobe turbidity sensors. OBSes deployed during this campaign were

15 calibrated in the laboratory using sediment obtained from the seabed near the measurement frames. However, due to the different response of optical sensors to sand versus other suspended particles, these calibrations may not be valid where high

concentrations of organic matter and flocculated fine sediment were present in the seawater. Further analysis is needed before the OBS measurements from areas with mixed sediments can be used.

## 7.3   Bathymetry and bed composition

The bed forms at one of the multibeam tracks is shown in Figure 8. Megaripples with a length of 6m are found in the channel (Profile 2), while the bed forms at the channel slope are much larger, approximately 20m (Profile 1).

Below all frames, small-scale ripples were present throughout the full measurement campaign. They were generally between 2 and 3.5 cm high, between 8 and 13 cm long and highly three-dimensional (Figure 8b).

The $d_{50}$ grain size at the lower shoreface ranges from around 200 μm around 12 m water depth to 230 μm (Ameland) and 300 μm (Terschelling) at a depth of around 20 m. The subsurface of the lower shoreface at Ameland and Terschelling is mostly sandy, with local clay bands of about 5 cm. The sediments consist of recent marine sands (Southern Bight Formation) on top of Holocene tidal deposits. Deeper cores from the database of the Dutch Geological Survey show Pleistocene sediments of the so-called Eem Formation below the tidal deposits.

In total 71 unique macrobenthic species were found at the ebb-tidal delta, mostly worms, crustacea, bivalves and echinodermata. On average, nine species per sample location were found.

## 8   Conclusions

A unique and comprehensive data set is presented, containing bathymetric data, hydrodynamic data, sediment data and benthic species distributions. The data was collected on the ebb-tidal delta of the Ameland Inlet and the lower shoreface offshore Ameland Inlet, Terschelling and Noordwijk, the Netherlands.

This dataset will help increasing the understanding of fundamental processes over complex bathymetries under the combined influence of waves, wind and tidal currents.

High-frequency hydrodynamic data was retrieved at several locations over the ebb-tidal delta, channels and lower shoreface. This high resolution dataset allows the analysis of intra-wave processes in this complex environment, including the influence of the tidal currents on wave transformation (de Wit et al., 2019). The measurement period was sufficiently long to capture several storm events but also calm conditions. This provides the opportunity to analyse the influence of wind, waves, and tidal flow on bed shear stresses, which are important for sediment transport.

The ebb-tidal delta has a complex bathymetry where hydrodynamic processes highly vary in space. The measurements at various locations and the X-band radar data can be used to analyse the spatial variation of flow, waves and sediment transport. The subsequent bathymetric surveys provide information about the morphological feedback resulting from these processes. First steps were made in Nederhoff et al. (2019) and Reniers et al. (2019) to calibrate numerical models of the area.

The combination of ecological and physical data can be used to develop and verify the (conceptual) models that describe interactions between biotic and abiotic processes.

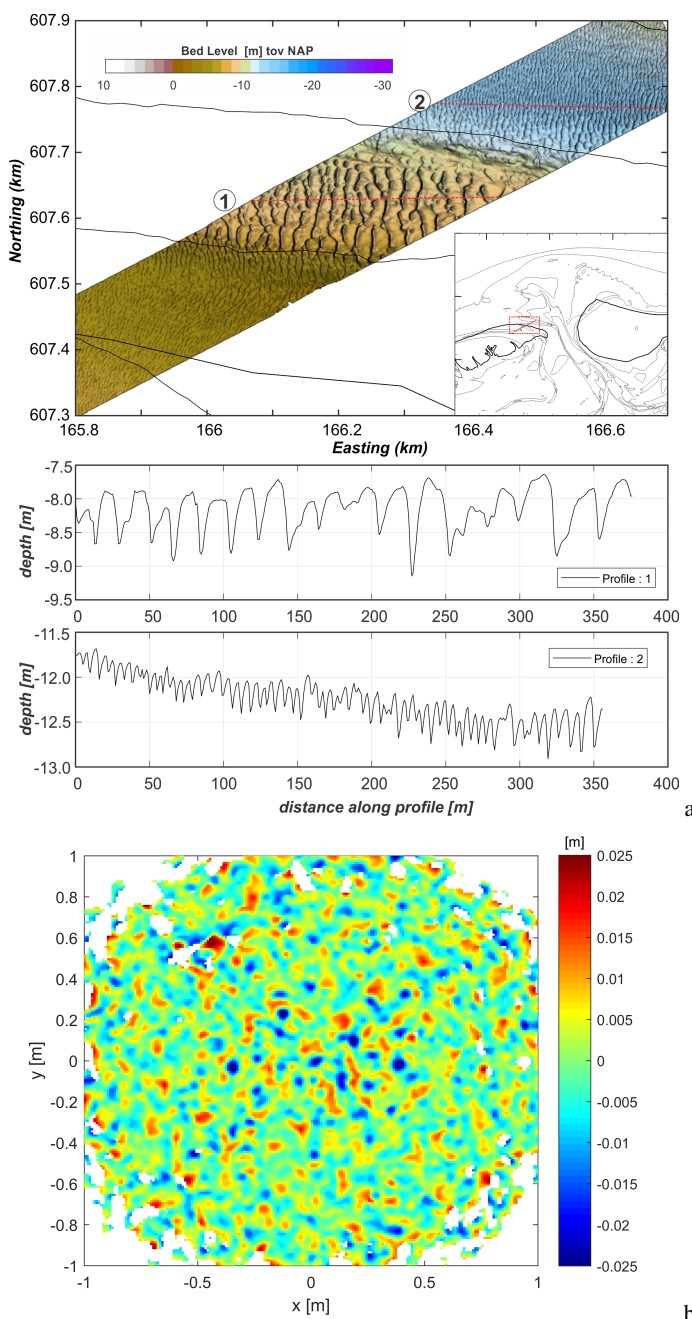

**Figure 8.** (a) Bed level as measured with multibeam for the band as indicated in lower right sub panel. The bed levels along two transects (as indicated in red with 1 and 2 in the map) are shown below the map. Clear variations in bed forms are visible, related to the larger scale bathymetry. (b) Relative bed level as measured below the frame with the sonar, indicating small-scale ripples.

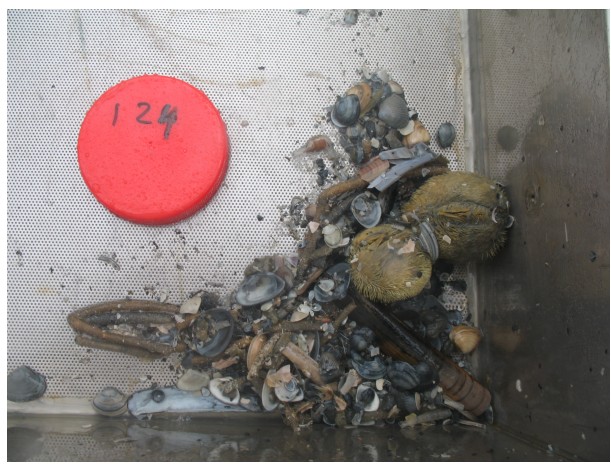

**Figure 9.** Example of benthos extracted from boxcore with 1 mm sieve, showing a.o. *Echinocardium cordatum* and *Ensis directus*.

The data can furthermore be used to improve algorithms for the modelling of the interaction between waves, flow and sediment transport. The high-frequency data is suitable for testing intra-wave scale models, where the spatial coverage allows comparison with larger scale wave resolving models.

This data is used within the joint SEAWAD and KustGenese2.0 program to analyze the Ameland ETD as representative of the other Wadden Sea ebb-tidal deltas, as well as the lower shoreface sediment dynamics. Ultimately, the results will be used for designing effective nourishment strategies to let the Dutch coastal zone keep pace with sea level rise.

*Author contributions.* BvP was involved in the set up, coordination and analysis; ZBW was initiator of SEAWAD and KustGenese2.0; MT, AR and FdW analysed ADCP/ADV/OBS frame data; SP analysed LISST/MPP frame data and grab samples; TV analysed boxcores and vibrocores; MG analysed the X-band radar; PKT and BG were involved in the set up of the campaign; HH and CS planned and carried out the boxcoring campaign; LB and MvdV analysed the sonar data; JvdW coordinated the 2nd and 3rd campaign; EE analysed multibeam data; FK and JWM were responsible for the operation of the field campaign; MvM took care of preparing the frames; JAA, PdV and GS carried out quality checks and processed data; KdH and RW were responsible for data storage the on the repository; HdL, CvGM coordinated KustGenese2.0.

*Competing interests.* No competing interests are present.

*Acknowledgements.* This work is part of the research programmes: KustGenese 2.0; and SEAWAD, a 'Collaboration Program Water' with project number 14489, which is financed by NWO Domain Applied and Engineering Sciences.

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
