# Peer review of "Measurements of Hydrodynamics, Sediment, Morphology and Benthos on Ameland Ebb-Tidal Delta and Lower Shoreface"

_Earth System Science Data, 2020_

## Referee Comment (RC1) · Neil Ganju (Referee) · 26 Feb 2020

This data release documents hydrodynamic and sediment transport measurements in areas of the Wadden Sea, landward and seaward of Ameland, and on the ebb tidal shoal. Overall, the report is clear, the data are accessible using accepted protocols (OpenDAP), and they will be of wide utility for coastal management locally and for basic sediment transport research globally. I have a few major suggestions, and a few minor comments.

Major comments:

It would be helpful to add a table of dates and coverages for the platforms/instruments, unless this is somewhere in the repository/supplemental that I didn't see.

Within the text, it would be good to document the pre-deployment calibration steps for the various instruments. As it stands the calibration for only one instrument is described (LISST), and it is embedded in the post-processing section. I would at least add documentation on calibration of the optical sensors, compass calibrations for the velocity meters, and pressure-zeroing for the pressure sensors. It may seem trivial but it is important for inter-operability to know how instruments were prepared.

Some of the terminology is unclear or inconsistent. For example, the term "tidal divides" is not a generally known term, and the use of "watersheds" to describe the landward drainage areas of the tidal channels is not the best choice. I think some time should be taken to tighten up some of these for clarity, mainly so the reader knows precisely where those ADCP measurements are being made without having to refer to the figure immediately. Perhaps the easiest terminology would be "tidal channels that drain the back-barrier basin" for the first use, and then "back-barrier tidal channels" after that?

Abstract: the abstract is awkwardly written. The details of the deployments probably don't need to be in the abstract, the reader will go to the map and table to decide if they can use the data. Suggest revising to something like this:

"The dataset obtained from the field campaign consists of: (i) single and multi-beam bathymetry; (ii) pressure, water velocity, wave statistics, sediment concentration, conductivity, temperature, and bedform morphology on the shoal; (iii) pressure and velocity at six back-barrier locations; (iv) bed composition and macro benthic species from box-cores and vibracores; (v) discharge measurements through the inlet; (vi) depth and velocity from X-band 10 radar; and (vii) meteorological data."

Minor comments:

P1, L10: "The synoptic nature of these measurements makes this dataset unique. . ."

[Figure]

P2, L9: "lose sediment across the offshore boundary" L13: "across the offshore boundary"

P3, L2: "Knowledge of these processes" L11: "gorge" is an unusual geomorphic term to use here L29-30: "reign free within its established boundaries" is unclear

P5, L3: upward looking RDI Monitors do not measure wave orbital motion L11: if turbidity was not calibrated to suspended-sediment concentration with water samples, then you should just say turbidity is used a proxy for SSC, but requires site-specific calibration.

P7, L14: "acoustic backscatter intensity", SPM not defined previously, spell out here and be consistent with use of suspended-sediment concentration vs. suspended particulate matter, suspended particle concentrations, etc. . .

P8, L1: "tidal divides" is not clear, and not defined previously. In the abstract, these locations were called "watersheds", and I suggested re-wording to "back-barrier". I suggest determining an optimal descriptor for these sites (back-barrier tidal channels?) and sticking with it throughout. L6: "Biannual bathymetric surveys" L28: what is a "laquer peal" or "lacquer peel"? L29: here and elsewhere, "vibracore" is the traditional spelling but if this is a brand name or regionally accepted spelling, OK.

P10, L25: I think it would be useful to have a section earlier in the documentation that identifies instrument calibration steps, like what is done with the LISST here. This could include the calibration of optical sensors, conductivity sensors, compass calibrations, zeroing-out the pressure, etc. This is pre-deployment info, whereas this section is post-processing. They should be handled separately for clarity if deemed important for this data release.

P12, L5: "in the North Sea"

Fig 8: is the sonar image from a location within the map in the upper panel? If so, show on map, if not then add an inset to show where it is.

Review by Neil Ganju, USGS

---

## Short Comment (SC1) · 11 Mar 2020

We highly appreciate the review. In the next version of the manuscript, we will implement the suggestions. We will at least include a table with dates of employment as well as an overview of the calibration of the instruments.

---

## Editor Comment (EC1) · Dirk Fleischer (Editor) · 14 May 2020

Dear Authors,

your manuscript describes a valuable dataset of sediment dynamics in the Netherland part of the Wadden Sea. The Data have been made available by the data repository of the TU Delft by CC-by licence. The repository is well chosen and the functionality provided to the user for accessing the data is very nice. This supports the mission of ESSD to make data reusable for future and extending research. Unfortunately, during checking the data in the repository a number of errors using the web frontend occurred and in the data file a number of "NaN" values occurred, which has not been mentioned

in the manuscript and was unexpected. Actually some files just contained "NaN" which is absolutely unacceptable for ESSD, as a data publication. If "NaN" occur this should be documented in the manuscript and only files should be included in the publication that actually contain reusable data. Please improve your data set description there is a AZG campaign mentioned, but the names of data files to mention other campaign acronyms. For future data users this is highly irritating and the manuscript is supposed to overcome these irritations to foster scientific progress. The abstract already should include information on the time periods and usability of the data. It would be good to have a table with data file descriptions, coverage and may be corresponding data files details within the manuscript. A minor comment: if available the impressive number physical samples should make use of persistent identifiers such as International Geo Sample Number (IGSN), but this only as a comment.

I provide here a detailed list of data files I had a closer look at:

2018_01_terschelling_dvt1/multiprobe_f3_dvt1.nc - temperature 100% NaN - conductivity 75% NaN - depth range 83-12800 - salinity 100% NaN - pH 100% NaN - turbidity 100% NaN - Chl 100% NaN - bga_pc 100% NaN - odo 100% NaN - time

2018_01_terschelling_dvt1/multiprobe_f4_dvt1.nc - temperature 100% NaN - conductivity 98% NaN - depth - salinity 100% NaN - pH 100% NaN - turbidity 100% Na - Chl 100% NaN - bga_pc 100% NaN - odo 100% NaN - time

2017_09_ameland_azg/multiprobe_f1_azg.nc - temperature - OK - conductivity - OK - depth - OK - salinity - OK - pH - OK - turbidity - OK - Chl - OK - bga_pc - OK - odo - OK - time

2017_09_ameland_azg/multiprobe_f3_azg.nc - temperature - OK - conductivity - OK - depth - negativ values? - salinity - OK - pH - OK - turbidity - OK - Chl - OK - bga_pc - OK - odo - OK - time

2017_09_ameland_azg/multiprobe_f4_azg.nc - temperature - OK - conductivity - OK -

depth - OK - salinity - OK - pH - OK - turbidity - OK - Chl - OK - bga_pc - values mainly 0.1 to 0.3, nothing >1 ? - odo - OK - time

2017_09_ameland_azg/multiprobe_f5_azg.nc - temperature 100% NaN - conductivity 95% NaN - depth - OK - salinity 100% NaN - pH 100% NaN - turbidity 100% NaN - Chl 100% NaN - bga_pc 100% NaN - odo 100% NaN - time

2017_11_ameland_dva/multiprobe_f1_dva.nc - temperature - OK - conductivity - OK - depth - OK - salinity - OK - pH - OK - turbidity - OK - Chl - OK - bga_pc - OK - odo - OK - time

2017_11_ameland_dva/multiprobe_f3_dva.nc - temperature - OK - conductivity - OK - depth - OK - salinity - OK - pH - OK - turbidity - OK - Chl - OK - bga_pc - OK - odo - OK - time

2017_11_ameland_dva/multiprobe_f4_dva.nc - temperature - OK - conductivity - OK - depth - OK - salinity - OK - pH - OK - turbidity - OK - Chl - OK - bga_pc - ver low variability of valus ? - odo - OK - time

2018_03_terschelling_dvt2/multiprobe_f1_dvt2.nc - temperature - OK - conductivity - OK - depth - OK - salinity - OK - pH - OK - Chl - OK - bga_pc - OK - odo - OK - time

2018_03_terschelling_dvt2/multiprobe_f3_dvt2.nc - temperature - OK - conductivity - OK - depth - OK - salinity - OK - pH - OK - turbidity - OK - Chl - OK - bga_pc - OK - odo - OK - time

2018_03_terschelling_dvt2/multiprobe_f4_dvt2.nc - temperature - OK - conductivity - OK - depth - OK - salinity - OK - pH - OK - turbidity - OK - Chl - OK - bga_pc - OK - odo - OK - time

2018_04_noordwijk_dvn/multiprobe_f1_dvn.nc - temperature - OK - conductivity - OK - depth - OK - salinity - OK - pH - OK - turbidity - OK - Chl - OK - bga_pc - OK - odo - OK - time

2018_04_noordwijk_dvn/multiprobe_f3_dvn.nc - temperature - OK - conductivity - OK - depth - OK - salinity - OK - pH - OK - turbidity - OK - Chl - OK - bga_pc - OK - odo - OK - time

2018_04_noordwijk_dvn/multiprobe_f4_dvn.nc - temperature 100% NaN - conductivity 99% NaN - depth - OK - salinity 100% NaN - pH 100% NaN - turbidity 100% NaN - Chl 100% NaN - bga_pc 100% NaN - odo 100% NaN - time

meteo_stations/knmi_meteo.nc - lon - OK - projection - OK - station - OK - time - OK (no time dependent measurement) - DD - OK - FH - OK - lat - OK

2018_04_noordwijk_dvn - adcp_hr_dvn_201804_F1P1.nc (repository returns Error message on html code = 403) - adcp_hr_dvn_201804_F3P2.nc (repository returns Error message on html code = 403) - adcp_hr_dvn_201804_F4P3.nc (repository returns Error message on html code = 403) - pressure_adcp_hr_dvn_201804_F1P1.nc (repository returns Error message on html code = 403) - pressure_adcp_hr_dvn_201804_F3P2.nc (repository returns Error message on html code = 403) - pressure_adcp_hr_dvn_201804_F4P3.nc (repository returns Error message on html code = 403) - temperature_adcp_hr_dvn_201804_F1P1.nc (repository returns Error message on html code = 403) - temperature_adcp_hr_dvn_201804_F3P2.nc (repository returns Error message on html code = 403) - temperature_adcp_hr_dvn_201804_F4P3.nc (repository returns Error message on html code = 403)

2018_01_terschelling_dvt1 - adcp_hr_dvt1_201801_F1P1.nc (repository returns Error message on html code = 403) - adcp_hr_dvt1_201801_F3P2.nc (repository returns Error message on html code = 403) - adcp_hr_dvt1_201801_F4P3.nc (repository returns Error message on html code = 403) - pressure_adcp_hr_dvt1_201801_F1P1.nc (repository returns Error message on html code = 403) - pressure_adcp_hr_dvt1_201801_F3P2.nc (repository returns Error message on html code = 403) - pressure_adcp_hr_dvt1_201801_F4P3.nc (repository returns Error message on html code = 403) - temperature_adcp_hr_dvt1_201801_F1P1.nc (repository returns Error message on html code = 403) - temperature_adcp_hr_dvt1_201801_F3P2.nc (repository returns Error message on html code = 403) - temperature_adcp_hr_dvt1_201801_F4P3.nc (repository returns Error message on html code = 403)

2018_03_terschelling_dvt2 - adcp_hr_dvt2_201803_F1P1.nc (repository returns Error message on html code = 403) - adcp_hr_dvt2_201803_F3P2.nc (repository returns Error message on html code = 403) - adcp_hr_dvt2_201803_F4P3.nc (repository returns Error message on html code = 403) - pressure_adcp_hr_dvt2_201803_F1P1.nc (repository returns Error message on html code = 403) - pressure_adcp_hr_dvt2_201803_F3P2.nc (repository returns Error message on html code = 403) - pressure_adcp_hr_dvt2_201803_F4P3.nc (repository returns Error message on html code = 403) - temperature_adcp_hr_dvt2_201803_F1P1.nc (repository returns Error message on html code = 403) - temperature_adcp_hr_dvt2_201803_F3P2.nc (repository returns Error message on html code = 403) - temperature_adcp_hr_dvt2_201803_F4P3.nc (repository returns Error message on html code = 403)v

---

## Author Comment (AC1) · 7 Sep 2020

Dear Dr. Ganju,

Thank you for your review and suggestions to improve the manuscript. Below, we reply to each point. You indicated one aspect that is still of our concern: the calibration of the OBS. We explain below what we did. Unfortunately, the concentration values cannot be provided with sufficient confidence.

Best regards,

[Figure]

Bram van Prooijen

————-

This data release documents hydrodynamic and sediment transport measurements in areas of the Wadden Sea, landward and seaward of Ameland, and on the ebb tidal shoal. Overall, the report is clear, the data are accessible using accepted protocols (OpenDAP), and they will be of wide utility for coastal management locally and for basic sediment transport research globally. I have a few major suggestions, and a few minor comments.

We highly appreciate that the reviewer considers the data set of wide utility.
Major comments:
It would be helpful to add a table of dates and coverages for the platforms/instruments, unless this is somewhere in the repository/supplemental that I didn't see.

We included a table with the frames for the different campaigns. See table 2 in the text. Furthermore, the site where the data is stored provides viewing options now as well. The position of the frames is directly visible on an interactive map, see for example Figure 1 or: https://data.4tu.nl/articles/dataset/KUSTGENESE2_0_ SEAWAD_Frame-Mounted_Velocity_Profiler/12705962

Within the text, it would be good to document the pre-deployment calibration steps for the various instruments. As it stands the calibration for only one instrument is described (LISST), and it is embedded in the post-processing section. I would at least add documentation on calibration of the optical sensors, compass calibrations for the velocity meters, and pressure-zeroing for the pressure sensors. It may seem trivial but it is important for inter-operability to know how instruments were prepared.
We agree that this aspect was not sufficiently described. Compass calibration for the instruments on the frames was carried out prior to the campaign. An example result

is shown below. We briefly explain the procedure in the text: "*The procedure for determining the offset of the compass was to rotate the mounting frame annotating every ten degrees the device heading angle (compass heading) and the true angle measured with high accuracy GPS (magnetic heading) not affected by the frame. This was repeated in reverse direction. An averaged compass deviation (from the two cycles) at a 10-degree interval was taken for the compass calibration.*"

OBS calibration is not straightforward. The OBS calibration was more problematic, as was discovered when we looked more closely at the full set of hydrodynamic and suspended particle measurements in the last year. A lab calibration was originally carried out with sediment from the bed. This sediment consisted almost solely of sand, which is reflective of the sediment composition across the ebb-tidal delta. An example of such a calibration is shown in Figure 3. During the campaign there are however strong indications that the suspended sediment contained fines advected from sources several kilometers away (see Pearson et al., 2019). These fines result in a significant response from the OBS at times when high suspended sand concentrations would not be expected (i.e. slack tide with few waves). The laboratory calibration with sand only (sediment from the bed) is therefore not representative. The OBS results are therefore limited to the voltage timeseries.

The following text is provided: "*A calibration was carried out at the laboratory of Utrecht University. Sediment (mainly sand) from the seabed was taken and mixed up in a mixing tank. The sediment concentration was slowly increased, obtaining the relation between voltage and sediment concentration. As the suspended sediment concentration in the field contained finesas well, the results from the laboratory (containing sand) cannot be translated directly into a concentration, see Pearson et al.(2019). Therefore, only the voltage output is provided.*"

The temperature, conductivity, pH, and DO sensors were calibrated prior to each campaign, following the procedures outlined by YSI Incorporated (2012). The sensors were placed in a container filled with water having known properties, and their readings were compared with the "true" values. The factory settings of the pressure sensors and velocity sensors in the ADVs and ADCPs were used.

Some of the terminology is unclear or inconsistent. For example, the term "tidal divides" is not a generally known term, and the use of "watersheds" to describe the landward drainage areas of the tidal channels is not the best choice. I think some time should be taken to tighten up some of these for clarity, mainly so the reader knows precisely where those ADCP measurements are being made without having to refer to the figure immediately. Perhaps the easiest terminology would be "tidal channels that drain the back-barrier basin" for the first use, and then "back-barrier tidal channels" after that?
The term "tidal divide" is often used to indicate the border of two different basins in the Wadden Sea. In Dutch, there is a special word for it: wantij. Searching scholar.google.com indeed indicates that tidal divides is mainly used by Dutch/German/Danish researchers. As watersheds is not the best choice either, we keep the definition "tidal divide". We first explain the location as the end of a tidal creek, and introduce the term tidal divide later on. We removed the term watershed.

Abstract: the abstract is awkwardly written. The details of the deployments probably don't need to be in the abstract, the reader will go to the map and table to decide if they can use the data. Suggest revising to something like this: "The dataset obtained from the field campaign consists of: (i) single and multi-beam bathymetry; (ii) pressure, water velocity, wave statistics, sediment concentration, con- ductivity, temperature, and bedform morphology on the shoal; (iii) pressure and velocity at six back-barrier loca- tions; (iv) bed composition and macro benthic species from box- cores and vibracores; (v) discharge measurements through the inlet; (vi) depth and velocity from X-band 10 radar; and (vii) meteorological data."
We agree and modified the abstract accordingly.

[Figure]

**Fig. 1.** interactive map with locations of the measurements.

[Figure]

**Fig. 2.** calibration of one of the compasses

[Figure]

**Fig. 3.** calibration curve for the OBS in the laboratory.

---

## Author Comment (AC2) · 7 Sep 2020

Dear Dr. Fleischer

Thank you for reviewing the paper. We replied to each comment/suggestion, see text
below.

Best regards,

Bram

[Figure]

Dear Authors, your manuscript describes a valuable dataset of sediment dynamics in the Netherland part of the Wadden Sea. The Data have been made available by the data repository of the TU Delft by CC-by licence. The repository is well chosen and the functionality provided to the user for accessing the data is very nice. This supports the mission of ESSD to make data reusable for future and extending research. Unfortunately, during checking the data in the repository a number of errors using the web frontend occurred and in the data file a number of "NaN" values occurred, which has not been mentioned in the manuscript and was unexpected. Actually some files just contained "NaN" which is absolutely unacceptable for ESSD, as a data publication. If "NaN" occur this should be documented in the manuscript and only files should be included in the publication that actually contain reusable data. Please improve your data set description there is a AZG campaign mentioned, but the names of data files to mention other campaign acronyms. For future data users this is highly irritating and the manuscript is supposed to overcome these irritations to foster scientific progress.

Some of the datasets indeed included only NaNs. This error was due to the inconsistent use of delimiters (i.e. ";" vs ",") in the output files for certain instruments, and was corrected in our post-processing. The corrected files have been uploaded to the repository. In addition to AZG, also the other acronyms have been indicated. This is or example done in the new table 2.

The abstract already should include information on the time periods and usability of the data.

The abstract now includes the years in which the measurements took place. "*The data has been obtained over the years 2017-2018. The most intensive campaign at the ETD of Ameland Inlet was in September 2017.*"
Extra sentences are added to explain how the data has been stored: "*The data sets are published in netCDF format and follow conventions for CF (Climate and Forecast) metadata. The data.4tu.nl site provides keyword searching options and maps with the*

*geographical position of the data.*"
Furthermore, it has been extended with a brief description of where the data set can be used for: "*The data provides opportunities to calibrate numerical models to a high level of detail. Furthermore, the data sets can be used for system comparison studies.*"

It would be good to have a table with data file descriptions, coverage and may be corresponding data files details within the manuscript.
Table 2 was added, indicating the periods and locations of the frames. As the data contains such a diverse amount of data, we could not make a single table to indicate the file descriptions and coverage. The data is stored within the data.4tu.nl site. This site contains many search options and (interactive) maps. We trust that these options are sufficient to find the data.

A minor comment: if available the impressive number physical samples should make use of persistent identifiers such as International Geo Sample Number (IGSN), but this only as a comment.
Our research group has a larger dataset of sediment samples for the Dutch Wadden Sea beyond just those presented in this paper and stored online here. IGSNs will be assigned to the dataset as a whole in the near future. In this way we hope to make our sediment sample catalogue more consistently organized across multiple field campaigns, both past and future.

---

## Author Comment (AC3) · 7 Sep 2020

see reply to other comments of Dr Fleischer.

———————————————————